# Potential SARS-CoV-2 Immune Correlates of Protection in Infection and Vaccine Immunization

**DOI:** 10.3390/pathogens10020138

**Published:** 2021-01-30

**Authors:** Yongjun Sui, Yonas Bekele, Jay A. Berzofsky

**Affiliations:** Vaccine Branch, Center of for Cancer Research, National Cancer Institute, NIH, Bethesda, MD 20892, USA; yonas.feyissa@nih.gov (Y.B.); berzofsj@mail.nih.gov (J.A.B.)

**Keywords:** SARS-CoV-2, neutralizing antibody, Th1 responses, T-cell immunity, innate immunity, type I interferon, trained immunity, mucosal immunity

## Abstract

Both SARS-CoV-2 infections and vaccines induce robust immune responses. Current data suggested that high neutralizing antibody titers with sustained Th1 responses might correlate with protection against viral transmission and disease development and severity. In addition, genetic and innate immune factors, including higher levels of type I interferons, as well as the induction of trained immunity and local mucosal immunity also contribute to lower risk of infection and amelioration of disease severity. The identification of immune correlates of protection will facilitate the development of effective vaccines and therapeutics strategies.

## 1. Introduction 

After crossing the species barrier, most likely from bats, severe acute respiratory syndrome coronavirus 2 (SARS-CoV-2) has recently emerged to infect humans and cause severe public health problems. Since the outbreak in late December 2019, more than 85.9 million cases and 1.86 million deaths have been reported worldwide as of 5 January 2021. Effective SARS-CoV-2 vaccines and therapeutic strategies are urgently needed. While some vaccines and monoclonal antibodies have been approved for clinical use, understanding their mechanism of protection can facilitate improvements that may become necessary as long-term efficacy data become available over the next few years.

SARS-CoV-2 infects target cells via angiotensin-converting enzyme 2 (ACE2) as the primary receptor and transmembrane protease serine 2 (TMPRSS2) as an activating protease [1,2]. In the respiratory system, since ACE2 and TMPRSS2 are expressed primarily in type II pneumocytes and a fraction of secretory cells [2,3,4], SARS-CoV-2 virus productively infects these target cells in the upper and lower respiratory tracts [5,6]. The virus also can infect endothelial cells of multiple organs such as lung, gut, liver and kidneys and result in damaging blood clots in multiple organs [1,2,7,8]. In humans, the infected patients displayed various COVID-19 disease symptoms, ranging from mild to severe pneumonia, and in some cases acute respiratory distress syndrome and lethal pulmonary failure [9,10,11]. Other COVID19-related pathological manifestations in gut, heart, and brain were also observed, and led to fatalities as well as some long-term debilitating sequelae in a fraction of survivors. The mechanisms determining the variable outcomes remain elusive. 

SARS-CoV-2 viral infections and vaccines induce robust immune responses, including innate and adaptive immune responses. However, the immune responses that can provide long-lasting protective immunity have yet to be fully determined. Here, we review the literature on what is currently known about the immune responses induced by SARS-CoV-2 infection and vaccines and focus on studies identifying potential immune correlates that predict protection. This is critical for development of vaccine and therapeutic strategies targeting SARS-CoV-2.

## 2. SARS-CoV-2 Infection-Induced Immunity That Mediates Protection against Disease

Natural SARS-CoV-2 infection results in both humoral and cellular immune responses. Humoral immunity induced by viral infection appears to play important roles in mediating protection against COVID-19 diseases. SARS-CoV-2 surface glycoproteins, mainly the spike protein, as well as the internal nucleocapsid protein, are the main targets of humoral immune responses. Most PCR-confirmed SARS-CoV-2–infected persons seroconverted by 2 weeks after disease onset [12]. Virus-specific IgG, IgA, and IgM responses were detected in acute and convalescent COVID-19 patients [13,14]. Zohar et al. explored the early evolution of the humoral response in COVID-19 patients and found complicated associations between disease severity and trajectories of antibody subtypes [15]. While a rapid and potent IgG class switching is correlated with survival, a delay but ultimate maturation of IgG subclasses is associated with milder disease [15]. Interestingly, such a trend was not found for IgA and IgM subtypes, which evolve rapidly regardless of disease severity [15].

The protective potential of humoral immunity was confirmed by some early studies on transfusion of plasma from convalescent patients or animal studies using potent purified monoclonal antibody cocktails [16,17,18,19,20]. However, a recent clinical trial did not verify the clinical benefits in patients with severe COVID-19 pneumonia who received convalescent plasma [21]. A short window of time for successful administration of the plasma/antibody cocktails might be the key to explain this discrepancy, as suggested also for administration of monoclonal antibodies (see below).

Similar to humoral immune responses, circulating SARS-CoV-2-specific CD4^+^ T-cell and CD8^+^ T-cell responses were detectable in most of SARS-CoV-2-infected patients within 1–2 weeks of symptom onset or at the convalescent stage [22,23]. Most virus specific CD4^+^T cell responses target spike and nucleocapsid glycoproteins as immunodominant. The responses were robust and mainly skewed towards Th1 cells, with the production of one or more of the Th1 cytokines, TNFα, IL-2, and IFNγ [22,23,24]. Though evidence of direct involvement of SARS-CoV-2-specific CD4^+^ T cells in mediating protection is lacking, for effective induction and long-term maintenance of antibody responses against COVID-19, a high-quality helper T cell response might be the key [25]. Several studies found that virus-specific CD4^+^ T cell responses were correlated with the magnitude of the anti-SARS-CoV-2 IgG and IgA against spike protein and nucleocapsid protein, live virus neutralizing antibody titers, and SARS-CoV-2 pseudovirus neutralization titers in COVID-19 patients [13,22,23,26].

Data on other coronavirus infections in animals and humans suggested that the humoral immune responses waned within a short period of time, whereas cellular immunity might persist longer. For SARS-CoV-2, while the duration of protective immunity is not yet known, a recent study suggests that IgG spike protein was relatively stable over at least 6 months, CD4^+^ and CD8^+^ T cell responses declined with a half-life of 3–5 months [27]. Previous exposure to other coronavirus might have cross-reactivity with SARS-CoV-2. In light of this, studies on pre-existing immunity against SARS-CoV-2 showed that cross-reactive T cells and antibodies are present [28,29,30]. While pre-existing T cell responses targeting a variety of viral proteins have been reported, the pre-existing antibody responses were predominantly of the IgG class and targeted the S2 subunit [28,29,30]. Whether the pre-existing immunity affects COVID-19 disease severity, or the dynamics of the current pandemic remains to be determined. Furthermore, SARS-CoV-2 -specific T cell responses were more sensitive to antigen exposure than antibody responses, as virus-specific T-cell responses were induced in antibody-seronegative individuals who had been exposed to SARS-CoV-2 [31,32].

SARS-CoV-2-specific CD8^+^ T cell responses were detectable in up to 70% of convalescent COVID-19 patients [13,22,23,28,31]. Virus-specific T cells displayed different phenotypes at different phases: highly activated cytotoxic phenotype at the acute stage, and polyfunctional and stem-like phenotype at the convalescent-stage [31]. Though there is some evidence suggesting that higher CD8^+^ T-cell responses were associated with mild disease, more studies are needed to elucidate whether virus-specific CD8^+^ T cells are pathogenic or protective in SARS-CoV-2 infections [23,31,33].

To identify the immune correlates of protection, though information from SARS-CoV-2 infections and vaccines are both valuable, extra caution is needed. Interpretation of the immune correlates from SARS-CoV-2–infected persons can be complicated and sometimes misleading. Like other viral infectious diseases, viral loads are a major driver of induced immune responses. Usually, patients with severe disease have high viral loads, which lead to high innate, and humoral/cellular immune responses, while the asymptomatic patients with lower viral loads induce lower immune responses. This is also true for SARS-Cov-2 infected patients. For example, vast numbers of studies have been carried out to characterize the humoral immune responses in COVID-19 acute or convalescent patients and most found that higher titers of neutralizing antibodies were induced in COVID-19 patients with severe disease than in those with mild disease [26]. The antibody responses also lasted longer in the patients with severe disease. Consistent with this, asymptomatic individuals manifested much weaker immune responses to SARS-CoV-2 infections than symptomatic ones [34]. Furthermore, in the early convalescent phase, IgG and neutralizing antibody levels in asymptomatic individuals declined much more quickly than those in the symptomatic patients [34]. However, these inverse correlations between disease severity and the magnitude of immune responses most likely indicate that severe disease, with high viral burden, leads to more robust immune responses, rather than the other way around.

## 3. Vaccine-Induced Immunity That Correlates with Protection against Viral Transmission and Disease

There are no fully licensed SARS-CoV-2 vaccines available so far. Multiple vaccine strategies, including adenovirus-vectored, inactivated virus, DNA-, mRNA-based platforms, and recombinant viral subunits/protein, have been utilized for SARS-CoV-2 vaccine development. So far, three vaccines (from Pfizer/BioNTech, Moderna, and AstraZeneca) have finished interim Phase III clinical trials, and obtained Emergency Use Authorization (EUA) in multiple countries. The immune responses have been measured and the vaccine efficacy has been tested in animal models and human clinical trials [35,36,37,38,39,40,41,42]. Most vaccine strategies induced humoral and cellular immune responses including virus-specific binding antibodies, neutralizing antibodies and virus-specific T-cell responses, in vaccinated macaques and human vaccinees [38,40,41,42,43,44,45,46,47]. Detailed information on the phase I-III clinical trials and the immune responses induced by these vaccines have been extensively reviewed [48,49,50]. Since most of studies have only immunogenicity data, the immune correlations that can be used to predict the outcome of viral infection and disease severity will not be ascertainable from these studies. Here we summarize the data from non-human primate (NHP) studies that had both immunogenicity and SARS-CoV-2 viral challenge outcomes presented (Table 1), in the hope to glean some clues on the immune correlates of protections.

Among these NHP studies, several of them identified immune parameters that were inversely correlated with viral loads and disease severity. Consistent with the fact that neutralizing antibody titer has been linked to protection against other pathogens and is considered a surrogate marker to predict viral clearance, two vaccine efficacy studies, one using DNA and another using Ad26 as delivery vectors, have demonstrated that vaccine-elicited neutralizing antibody titer correlated with protective efficacy, and thus suggested neutralizing antibody as an immune correlate of protection [36,39]. In the DNA vaccine study, Yu et al. found that besides neutralizing antibody, Spike-and receptor-binding domain (RBD)-specific antibody-dependent complement deposition (ADCD) responses inversely correlated with viral loads [39]. Specifically, RBD-specific FcγR2a-1 binding antibody mediating ADCD responses and neutralizing antibody (Nab) titers of IgG2 antibodies improve the correlation in two nonlinear regression analysis models [39]. In SARS-CoV-2 infection, antibodies against viral protein, especially the RBD of the spike protein, can neutralize the virus by preventing the virus from binding to the ACE2 receptor on susceptible cells, and thus block viral entry, transmission and infections [52]. In another study using Ad26 as the vaccine platform, Mercado et al. found that virus-specific serum antibody titers, including ELISA titers, pseudovirus neutralization titers and live virus neutralization titer, inversely correlated with peak viral load in bronchoalveolar lavage and nasal swabs [36]. Other antibody features such as FcγR2a-3 or IgM-mediated ADCD responses increased the correlation with protective efficacy, suggesting that in addition to neutralizing antibodies, other binding and functional antibodies may play roles as well [36]. Moreover, neutralization activity in the serum of the mRNA-1273 vaccinated macaques was associated with quick clearance of the input virus in the nasal secretions [38]. Indeed, in the Phase 3 clinical studies of ChAdOx1 [53,54], BNT162b2 [55], and mRNA-1273 [56], where high titers of neutralizing antibody responses were induced, 70–95% of protective efficacy was achieved, suggesting neutralizing antibody titer as a correlation marker. Further confirmation studies are needed to compare the immune responses in the viral infected vs. uninfected vaccinees to answer the questions whether neutralizing antibody responses alone are sufficient, or whether another biomarker will also predict protection.

In both DNA and adeno26-vectored studies, virus-specific CD4^+^ T cell and CD8^+^ T cell responses in blood were detectable; however, neither of them correlated with protection [36,39]. Another mouse study conducted by Yang et al. showed that a spike subunit vaccine mediated protection against SARS-CoV-2 infection [57]. When adoptive transfer of immune sera vs. splenic T cells from the vaccinated mice was conducted, adoptive transfer of splenic T cells (CD4^+^ and CD8^+^ cells) did not protect the human ACE2-transgenic mice from SARS-CoV-2 challenge, whereas transfer of sera from the immunized mice did [57]. However, a recent CD8 depletion study in the macaque model clearly demonstrated that cellular immunity, especially CD8^+^ T cell responses, played a pivotal role in protecting against viral rechallenge with SARS-CoV-2 in convalescent macaques with waning antibody titers [58]. In this study, the authors also found that adoptive transfer of relatively low titer purified polyclonal IgG from convalescent macaques could protect naïve macaques against SARS-CoV-2 rechallenge. Using logistic regression models, pseudovirus NAb titers of 50, RBD ELISA titers of 100, and S ELISA titers of 400, were determined to be the antibody titer thresholds required for protection.

Even if no direct association with protection has been identified, virus-specific T cell responses, especially Th1 and T follicular helper (TFH) responses, are pivotal for the induction, and more importantly, the functionality, the isotype, and the maintenance of antibody responses. Indeed, data suggested that protective immunity was associated with coordinated virus-specific adaptive immune responses, including neutralizing antibody responses, and CD4^+^ and CD8^+^ T cells [59]. Future animal studies with specific depletion of certain arms of immune responses will facilitate the elucidation of the protective mechanisms.

## 4. Innate Immunity and Trained Immunity Contribute to Protection

Innate immune responses participate in protecting against SARS-CoV-2 infection. Two recent studies demonstrated that inborn genetic mutations of type I interferon and autoantibodies blocking type I interferon are associated with severe COVID-19 disease, suggesting that type I interferons are key protective factors to prevent severe disease [60,61]. These findings also provide a potential explanation for the wide variety of clinical disease phenotypes. The fact that these two types of flawed interferon responses underlie 14% of COVID-19 severe cases demonstrates the important contribution of innate immunity in preventing COVID-19 disease [62]. Conversely, overproduction of inflammatory cytokines, which may be coupled with low type I IFN response, has been found to exacerbate SARS-Cov-2 disease. Laing et al. have characterized an innate immune signature that could be used to guide clinical care and treatment. The signature includes interleukin-10 and interleukin-6, which could anticipate subsequent clinical progression [63]. In severe COVID-19 patients, a systemic pro-inflammatory signature, including elevated plasma IL-6 and C-reactive protein (CRP) levels, was associated with clinical worsening and 2-month mortality [64].

A recent genome-wide association study identified several host genes that showed significant associations with severe COVID-19 disease [65]. Low expression of interferon receptor gene IFNAR2, and high expression of tyrosine kinase 2, were found to be linked to life-threatening disease. High expression of the monocyte/macrophage chemotactic receptor CCR2 is also associated with severe COVID-19. Moreover, gene cluster/gene encoding antiviral restriction enzyme activators (OAS1, OAS2, OAS3), and dipeptidyl peptidase 9 (DPP9) also were linked to disease severity and might be involved in affecting the disease outcome. Brunchez et al. found that major histocompatibility complex (MHC) class II transactivator (CIITA) and CD74 can inhibit SARS-CoV-2 viral entry by blocking the cathepsin pathway [66].

Trained immunity, first suggested by Netea, refers to enhanced responsiveness by myeloid cells and natural killer cells when they reencounter pathogens [67]. In the era of COVID-19, studies found that in countries where Bacillus Calmette–Guérin (BCG) vaccination is widely given, a lower infection rate and COVID-19-related mortality rate were observed [68,69]. Thus, the hypothesis that some pathogens or vaccines such as BCG might confer protection against SARS-CoV-2 was proposed. Though some studies did not find evidence that BCG protects against SARS-CoV-2 infection and COVID19 disease [70,71], others did. Escoba et al. conducted an epidemiological study to assess whether BCG vaccination was associated with COVID-19 mortality. After mitigating multiple confounding factors, they observed that every 10% increase in the BCG vaccination prevalence led to a 10.4% reduction in COVID-19 mortality [72]. Another study from a diverse cohort of health care workers found that decreased SARS-CoV-2 IgG-positive rates and reduced COVID-19-related clinical symptoms were associated with BCG vaccination [73]. While a prospective clinical trial on BCG vaccination for prevention and amelioration of COVID-19 severity (NCT04534803) is ongoing, the mechanism by which BCG vaccination affects the severity of COVID-19 has yet to be determined.

## 5. Mucosal Immunity Protects Viral Transmission at the Frontlines

Mucosal immunity is important for COVID-19 because the virus infects and is transmitted primarily through the upper and lower respiratory tracts, where the ACE2 receptors for the virus are present. By inducing mucosal antibody and T cell immunity, mucosal vaccines and therapeutic strategies are expected to be able to abort infection locally at an earlier stage at the site of transmission before the virus disseminates systemically.

A recent study using a human ACE2-expressing mouse model showed that an intranasal ChAd-SARS-CoV-2 vaccine induced high levels of neutralizing antibodies and promoted systemic and mucosal IgA and T cell responses, which led to almost full protection against viral transmission at the upper and lower respiratory tract. The fact that the intranasal vaccine was better than the intramuscular counterpart in preventing upper and lower respiratory tract infection demonstrated the importance of mucosal immunity in mediating protection against SARS-CoV-2 infections [74]. Indeed, systemic vaccines such as mRNA and adenovirus vaccines provided only limited protection against input viral clearance and resulted in persistent viral presence in nasal swabs [36,38,43]. Prospectively, mucosal vaccines were proposed to be better at clearing the virus in the airways, and thus preventing virus transmission [50,75,76]. A mucosal vaccine which is in development in our lab showed more effective clearance of nasal input virus than a systemic vaccine (unpublished data).

## 6. Conclusions

Overall, as summarized in Table 2, the current view of an optimal immune response usually consists of high titers of neutralizing antibodies with durable Th1-biased T cells. Antibody repertoire, especially the functionality of IgG, including Fc receptor binding, Fc effector activity, and complement-fixation activity, played an important role in mediating protection. Favorable innate immunity, trained immunity, and mucosal immunity also contribute substantially to optimal protection against SARS-CoV-2 viral transmission and severe COVID19 disease development. Conversely, it is possible that different vaccine platforms may have different protective mechanisms, and therefore have different immune correlates of protection. As current vaccines may require additional boosts to maintain long-term immunity, a better understanding of which immune responses are most effective at preventing transmission and controlling infection to prevent severe disease may be critical to design the best boosts and adjuvants that can determine the type of response. For example, mucosal boosts may be more effective at reducing virus in the nasal cavity and thus in reducing transmission to others [50] (Sui et al., manuscript submitted). Moreover, adjuvants can skew the isotype of antibodies induced, as well as the cytokine profile of T cells elicited [77,78,79]. With the release of more data on phase III clinical trials from different vaccines, more immune correlates of protections will be identified in the future.

## Figures and Tables

**Table 1 pathogens-10-00138-t001:** The correlations between vaccine-induced immune responses and viral loads in macaque model after SARS-CoV-2 viral challenges.

Vaccines	Manufacturer/Location	EUA	Antigen	Delivery Vectors	Ab-Pseudo/Live Nab *	T Cell-CD4^+^/CD8^+^T	Challenge Doses	Protection in BAL	Protection in Nasal Tissue	Refs
ChAdOx1 nCov-19	AstraZeneca/UK	UK/Argentina/India/Mexico/Braizil etc.	FL native S	# RI chimp Ad type E	10-160/NA	detectable in both	2.6 × 10^6^ TCID50	75% (3/4, prime, 3/4, boost)	0%	[43]
Ad26. COV2.S	Johnson&Johnson/USA		FL 2P stabilized S	RI human Ad26 type D	408/113	Th1/detectable	1.1 × 10^4^ PFU	100% (6/6)	87% (5/6)	[36]
mRNA-1273	Moderna/USA	USA/Canada/EU/UK etc.	FL 2P stabilized S	mRNA in lipid nanoparticle	1862/3481	Th1 and Tfh/low	7.6 × 10^5^ PFU	87% (7/8)	100% (8/8)	[38]
Sinopharm (BBIBP-CorV)	Sinopharm/China	China/UAE/Pakistan/Bahrain/Kuwait/Egypt/Brazil etc.	Inactivated whole virus	AIOH	NA/200	NA/NA	1 × 10^6^ TCID50	Throat: 4 log reduction (day5)		[51]
PiCovacc	Sinovac/China	China/Indonisia etc.	Inactivated whole virus	AI(OH)3	NA/10-100	NA/NA	1 × 10^6^ TCID50	Throat: 3 log reduction (day5)		[35]
DNA			variants of S protein		NA/74	Th1/detectable	1.1 × 10^4^ PFU	3.1 (Bal) and 3.7 (NS) log reduction		[39]

*: Nab: neutralizing antibody; # RI: replication-incompetent. TCID50: 50% of Tissue Culture Infectious Dose; EUA: Emergency Use Authorization; UK: United Kingdom; EU: European Union; UAE: United Arab Emirates; FL: full length; BAL: Bronchoalveolar lavage; NS: nasal swab; PFU: plaque-forming unit; NA: none.

**Table 2 pathogens-10-00138-t002:** Immune biomarkers that predict the outcome of SARS-CoV-2 infections.

Biomarkers		Roles in Viral Infections	References
Humoral immunity	Neutralizing antibodies S-, RBD- binding antibodies	Correlated with protective efficacy	[16,17,18,19,20,36,39]
	Antibody repertoire: subtypes, Fc receptor binding, Fc effector activity, complement-fixation activity	Contribute to protection	[15,36,39]
Cellular immunity	CD4+T helper cells	Effective induction and long-term maintenance of antibody responses	[13,22,23,26]
	CD8+T cell responses	Higher frequency associated with milder disease; partially participating in viral control	[23,31,33,58]
Innate immunity	Type I IFNs	Key protective factors to prevent severe disease	[60,61]
	Pro-inflammatory cytokines	Exacerbate COVID-19 diseases	[63,64]
	Host genes: OAS1, OAS2, OAS3	Antiviral restriction enzyme activators	[65]
	Low expression of IFNAR2, and high expression of TYK2	Causal link to life-threatening disease	[65]
	High expression of CCR2	Associated with severe Covid-19	[65]
	(MHC) class II transactivator (CIITA) and CD74	Block cathepsin pathway to inhibit SARS-CoV-2 viral entry	[66]
Trained immunity	BCG vaccination	Decreased SARS-CoV-2 IgG-positive rates and reduced COVID-19-related clinical symptoms	[72,73]
	BCG vaccination	No correlations	[70,71]
Mucosal immunity	Intranasal vaccine	Better than the intramuscular counterpart in preventing upper and lower respiratory tract infections	[74]

RBD: receptor-binding receptor; MHC: major histocompatibility complex; BCG: Bacillus Calmette–Guérin.

## Data Availability

Not applicable.

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
