# Peer review of "Potential SARS-CoV-2 Immune Correlates of Protection in Infection and Vaccine Immunization"

_pathogens, 2021, doi:10.3390/pathogens10020138_

Round 1

Reviewer 1 Report

The review article is well-written and clear. Please find some minor comments below:

  1. Pg 2, 1st paragraph, lines 56-57: the use of the word “convergence” to describe the evolution of IgG response, as described in ref 15, seems inappropriate. It may be clearer to use “maturation”, “development” or “evolution” of the IgG "repertoire” or “subclasses”.
  2. Table 1 : please define abbreviation “NA”.
  3. Table 1 and related text: The authors ought to include the article by McMahan et al. (Correlates of protection against SARS-CoV-2 in rhesus macaques. Nature, 2020; DOI: 1038/s41586-020-03041-6) where they found that adoptive transfer of purified IgG from convalescent rhesus macaques protected naive recipient macaques against challenge with SARS-CoV-2 in a dose-dependent fashion. Depletion of CD8+T cells in convalescent macaques partially abrogated the protective efficacy of natural immunity against re-challenge with SARS-CoV-2.
  4. Humoral response: other than high neutralizing antibody levels, several references (such as 15, 39) also mention the importance of the IgG repertoire in mediating protection, in particular, RBD-specific FC(gamma)R binding antibodies and complement-fixing antibodies. This information should be highlighted in the abstract and/or conclusion.
  5. Pro-inflammatory response: it has been reported and discussed by researchers that overproduction of inflammatory cytokines (coupled with low type I IFN response) exacerbate SARS-Cov-2 disease. The authors ought to touch upon this point.

Author Response

The review article is well-written and clear. Please find some minor comments below:

  1. Pg 2, 1stparagraph, lines 56-57: the use of the word “convergence” to describe the evolution of IgG response, as described in ref 15, seems inappropriate. It may be clearer to use “maturation”, “development” or “evolution” of the IgG "repertoire” or “subclasses”.

Response: We have changed “convergence of IgG” to “the maturation of IgG subclasses”

  1. Table 1 : please define abbreviation “NA”.

Response: We have added “NA: none” in Table 1.

  1. Table 1 and related text: The authors ought to include the article by McMahan et al. (Correlates of protection against SARS-CoV-2 in rhesus macaques. Nature, 2020; DOI: 1038/s41586-020-03041-6) where they found that adoptive transfer of purified IgG from convalescent rhesus macaques protected naive recipient macaques against challenge with SARS-CoV-2 in a dose-dependent fashion. Depletion of CD8+T cells in convalescent macaques partially abrogated the protective efficacy of natural immunity against re-challenge with SARS-CoV-2.

Response: We would like to thank the review to refer this article, which has identified/confirmed the protective role of IgG antibody, and CD8+T cells against SARS-Cov-2. Since this article focused on natural immunity, and did not use vaccine, we therefore did not include it in Table1. However, we revised the manuscript by adding the following text on page 5, line184-191. 

“However, a recent CD8 depletion study in the macaque model clearly demonstrated that cellular immunity, especially CD8+ T cell responses, played a pivotal role in protecting against viral rechallenge with SARS-CoV-2 in convalescent macaques with waning antibody titers {McMahan, 2020 #93}. In this study, the authors also found that adoptive transfer of relatively low titer purified polyclonal IgG from convalescent macaques could protect naïve macaques against SARS-CoV-2 rechallenge. Using logistic regression models, pseudovirus NAb titers of 50, RBD ELISA titers of 100, and S ELISA titers of 400, were determined to be the antibody titer thresholds required for protection.”

  1. Humoral response: other than high neutralizing antibody levels, several references (such as 15, 39) also mention the importance of the IgG repertoire in mediating protection, in particular, RBD-specific FC(gamma)R binding antibodies and complement-fixing antibodies. This information should be highlighted in the abstract and/or conclusion.

Response: We added in the conclusion the following on page 6 line 264-266.

“Antibody repertoire, especially the functionality of IgG, including Fc receptor binding, Fc effector activity, and complement-fixation activity, played an important role in mediating protection.”

  1. Pro-inflammatory response: it has been reported and discussed by researchers that overproduction of inflammatory cytokines (coupled with low type I IFN response) exacerbate SARS-Cov-2 disease. The authors ought to touch upon this point.

Response: We have revised the manuscript to address the pro-inflammatory responses on page 5 line 208-214.

“Conversely, overproduction of inflammatory cytokines, which may be coupled with low type I IFN response, has been found to exacerbate SARS-Cov-2 disease. Laing et al. have characterized an innate immune signature that could be used to guide clinical care and treatment. The signature includes interleukin-10 and interleukin-6, which could anticipate subsequent clinical progression [59]. In severe COVID-19 patients, a systemic pro-inflammatory signature, including elevated plasma IL-6 and CRP levels, was associated with clinical worsening and 2-month mortality [60]. ”

Reviewer 2 Report

Overall well written.

Author Response

Thank you!

Reviewer 3 Report

This manuscript by Youngjun Sui et al. describe very intersting data on SARSCoV2-related immunity and vaccine immunization.

This subject is already extensively covered in literature and the manuscript deserve actualization from human data (more recent that NHP).

Moreover, some modification could be needed.

"et al." : write a "." and in italic.

Table 1 : Manufacturer's name/location and countries that use this vaccine could be usefull

A table is needed for the reader to easily summarize and understand which biomarkers are associated with better or worse outcome.

Author Response

This manuscript by Youngjun Sui et al. describe very intersting data on SARSCoV2-related immunity and vaccine immunization.

1. This subject is already extensively covered in literature and the manuscript deserve actualization from human data (more recent that NHP).

Response: We agree with the reviewer that the topics on SARSCoV2-related immunity and vaccine immunization have been extensively reviewed in literature. Both SARS-CoV-2 infection and vaccination induce robust immunity, yet only a portion of them could be biomarkers to predict the infection outcome. In this manuscript, we focus on the papers that have identified or tried to identify such biomarkers.  So far, vast body of literature of human clinical trials on SARS-CoV-2 vaccines have been published, in the Phase 3 clinical studies, which have achieved 70-95% of protective efficacy, high titers of neutralizing antibody responses were induced, and most likely correlated with protection.  By further comparing the immune responses in the viral infected vs. uninfected vaccinees, we would expect that such biomarkers will be available in the future. We have added the following on page 4 line 170-176:

“Indeed, in the Phase 3 clinical studies of ChAdOx1[53,54], BNT162b2 [55], and mRNA-1273 [56], where high titers of neutralizing antibody responses were induced, 70-95% protective efficacy was achieved, suggesting neutralizing antibody titer as a correlation marker. Further confirmation studies are needed to compare the immune responses in the viral infected vs. uninfected vaccinees to answer the questions whether neutralizing antibody responses alone are sufficient, or another biomarker will also predict protection.”

2. Moreover, some modification could be needed.

"et al." : write a "." and in italic.

Response: We have revised accordingly.

3. Table 1 : Manufacturer's name/location and countries that use this vaccine could be usefull

Response: We have added the information in the table 1.

4. A table is needed for the reader to easily summarize and understand which biomarkers are associated with better or worse outcome.

Response: We would like to thank the reviewer for this excellent suggestion. We have revised the manuscript by adding a new Table 2 to summarize the biomarkers involved.

Round 2

Reviewer 3 Report

The authors have taken into account my previous comments and, I think, deserve publication in the present form.